# Advances in Semiconductor-Based Nanocomposite Photo(electro)catalysts for Nitrogen Reduction to Ammonia

**DOI:** 10.3390/molecules28062666

**Published:** 2023-03-15

**Authors:** Cheng Zuo, Qian Su

**Affiliations:** College of Chemical Engineering and Environmental Engineering, Weifang University, Weifang 261061, China

**Keywords:** photocatalyst, electrocatalyst, nitrogen reduction, nanomaterials

## Abstract

Photo(electro)catalytic nitrogen fixation technology is a promising ammonia synthesis technology using clean solar and electric energy as the driving energy. Abundant nitrogen and water as raw materials uphold the principle of green and sustainable development. However, the generally low efficiency of the nitrogen reduction reaction has seriously restricted the application and development of this technology. The paper introduces the nitrogen reduction process and discusses the main challenges and differences in the current photo(electro)catalytic nitrogen fixation systems. It focuses on promoting the adsorption and activation of N_2_ and the resolution and diffusion of NH_3_ generated. In recent years, reviews of the modification strategies of semiconductor materials in light of the typical cases of nitrogen fixation have been reported in the literature. Finally, the future development trend of this field is analyzed and prospected.

## 1. Introduction

Ammonia is a carbon-free hydrogen storage compound widely used in agriculture, the chemical industry, medicine, energy storage, and other fields [1]. The ammonia Haber–Bosch (HB) synthesis process is undoubtedly one of the most important inventions in modern history and is also the most mature process for the large-scale production of nitrogen-based fertilizer. The annual output of NH_3_ reaches 160 million tons, of which 80% is used to produce fertilizer to feed more than 70% of the world’s population. The HB process uses an iron-doped catalyst to convert N_2_ and H_2_ mixed in a 1:3 ratio to ammonia. This process requires reaction conditions of high temperature (400–600 °C) and high pressure (200–400 atm) to activate the highly stable N≡N (bond dissociation energy up to 941 kJ·mol^−1^) in the N_2_ molecule. However, the HB process consumes about 2% of the world’s total energy annually and emits 300 million tons of CO_2_ greenhouse gas, accounting for about 1.6% of global emissions [2]. Therefore, to alleviate energy consumption and environmental problems, seeking a sustainable and cost-effective ammonia synthesis strategy is necessary. NH_3_ synthesis strategies such as biocatalysis, electrocatalysis, and photocatalysis have been studied on a laboratory scale. Among them, photo(electro)catalytic nitrogen reduction synthetic ammonia technology, driven by inexhaustible solar energy, reduces nitrogen to ammonia with high energy density, easy storage and transportation. In addition, the nitrogen reduction process can also be carried out under environmental conditions while achieving zero carbon emissions, which is harmless to the environment. It is considered a potential alternative to the industrial HB process to generate NH_3_, which has aroused the keen attention of society.

Although the Gibbs free energy of ammonia synthesis is negative, it cannot spontaneously react at room temperature and environmental pressure due to the stability and chemical inertness of N_2_. It is an arduous task to convert N_2_ into NH_3_ directly. In 1977, Schrauzer and Guth [3] found that Fe_2_O_3_-doped TiO_2_ powder could reduce nitrogen to ammonia under UV light irradiation. Subsequently, it was reported [4] that desert sands formed after weathering titanite-rich rocks could fix nitrogen by photoreaction in 1983. Tennakone’s group [5] prepared ultrafine Fe(O)OH particles with photocatalytic activity in reducing nitrogen to ammonia and discussed the preparation method, ammonia yield, and reaction mechanism of the catalyst in 1991. These landmark events confirmed the possibility of using solar energy to reduce nitrogen to synthesize ammonia directly and raised the curtain on the exploration and application of semiconductor materials in the field of photocatalytic nitrogen fixation. However, the heterogeneous reaction efficiency in an aqueous solution is low due to the weak adsorption and the activation difficulty of N_2_ molecules on the catalyst surface, the participation of high-energy intermediates, and the complex multi-electron transfer reaction pathway. In addition, the photocatalytic reduction reaction is affected by the easy recombination of photogenerated electron holes and the weak reduction ability of photogenerated electrons. The reaction efficiency of electrocatalysis is limited by high overpotential, low current density, low selectivity, and competition for hydrogen evolution reactions. Therefore, developing highly active and stable catalysts is vital in converting N_2_ to NH_3_.

Photo(electro)catalysis has developed rapidly in the past decades, and many different types of photo(electro)catalysts have been proposed, including noble metals, metal complexes, organic molecules, ions, and semiconductors. Since semiconductors offer significant advantages in manufacturing cost, material toxicity and stability compared to other situations, they show attractive promise in photocatalysis research and have long been studied and tested. So far, researchers have designed and developed various semiconductor photo(electro)catalytic materials, including metal oxides (TiO_2_, Fe_2_O_3_, ZnO) [6,7,8], metal sulfides (MoS_2_, CdS, ZnS) [9,10,11,12], bismuth-based materials (BiOX, X = Cl, Br, I) [13,14], carbon-based materials (diamond, graphene, g–C_3_N_4_) and MOFs. Some progress has been made in the design optimization of semiconductor materials and their photocatalytic nitrogen fixation performance [15]. However, the overall reaction efficiency of the photo(electro)catalytic nitrogen fixation was still very low. It did not meet the requirements of practical applications, so there was an urgent need for breakthroughs in the knowledge and understanding of fundamental scientific issues. This paper introduces the process and mechanism of nitrogen reduction and points out the main challenges of the photo(electro)catalytic nitrogen fixation system. It focuses on the modification of semiconductor nanomaterials and proposes strategies to enhance the performance of photo(electro)catalytic nitrogen fixation, summarizing the current status of recent research on nitrogen reduction. Finally, future research will provide new ideas and inspiration for the photo(electro)catalytic nitrogen fixation process under mild conditions.

## 2. Nitrogen Reduction Reaction

Semiconductors generate photogenerated electrons (*e*^−^) and photogenerated holes (*h*^+^) under UV and visible light excitation. The photogenerated electrons on the conduction band (CB) reduce N_2_ to NH_3_ (N_2_ + 6H^+^ + 6*e*^−^ → 2NH_3_). The photogenerated holes on the valence band (VB) oxidize H_2_O to O_2_ and provide protons for the reduction reaction (3H_2_O + 6*h*^+^ → 6H^+^ + 1.5O_2_), achieving low energy consumption and zero carbon emissions for the nitrogen reduction reaction (NRR) (Figure 1) [16].

Thermodynamically, whether the photocatalytic reaction could occur depends on the redox potential of the adsorbate and the position of the energy band of the semiconductor. In the nitrogen reduction reaction, the semiconductor’s conduction band position should be higher (more negative) than the nitrogen hydrogenation reduction potential, while the valence band should be lower (more positive) than the oxygen precipitation potential. Under the combined effect of light and catalyst, N_2_ is reduced to NH_3_ by a series of photogenerated electrons and water-born proton injection; the hydrogenation reaction in the process with the corresponding reduction potential [17] was shown in Table 1. The first electron transfer (Equation (1c)) and proton-coupled electron transfer (Equation (1d)) processes in the hydrogenation process were high-energy transition states with slow reaction kinetics that reduced the overall reaction efficiency.

The exact mechanism of the nitrogen reduction reaction was ambiguous. Currently, there are three recognized catalytic hydrogenation mechanisms: dissociation mechanism, association mechanism and enzyme catalytic mechanism (Figure 2) [18]. The dissociation mechanism refers to the direct breaking of the chemical bond in the N_2_ molecule and the formation of two isolated nitrogen atoms on the catalyst surface, which then undergoes hydrogenation to produce NH_3_. The H-B process proceeds through this mechanism, breaking the N≡N bond before adding hydrogen atoms. It explains the necessity of high temperature and high-pressure reaction conditions in the H–B process. The photocatalytic or electrocatalytic nitrogen fixation process was more in line with the association mechanism by following the principle of low energy consumption, where N_2_ molecules were hydrogenated continuously without breaking the N≡N triple bond until the first NH_3_ molecule was released. Depending on the hydrogenation mode, the association mechanism was further divided into two types: the distal binding mechanism and the alternate binding mechanism. In the distal binding mechanism, protonation occurs preferentially on the nitrogen atoms far from the catalyst surface without direct interaction with the catalyst and after releasing the first NH_3_ molecule. The remaining adsorbed nitrogen atoms continued hydrogenating until the second NH_3_ molecule was produced. In the alternating binding mechanism, the two nitrogen atoms were converted to NH_3_ by accepting protons in turn, and the synthesized NH_3_ molecule was desorbed from the catalyst surface by a final N–N break. The enzyme-catalyzed mechanism applies to hydrogenation reactions occurring with nitrogen-fixing enzymes and other catalysts, in which both N atoms are bound to the nitrogen-fixing enzyme or catalyst, and hydrogenation coincides [19,20,21]. In summary, the photocatalytic nitrogen reaction was summarized as three steps: (1) N_2_ molecules were adsorbed on the reaction sites of the catalyst; (2) electrons and protons were obtained through a proton–coupled electron transfer process, and the N≡N bond was broken to produce NH_3_; (3) NH_3_ molecules were desorbed from the catalyst surface. Understanding the binding mode of N_2_ molecules on the photocatalyst was essential for studying the reaction mechanism of nitrogen reduction.

## 3. Challenges of NRR

Due to the inert nature of N_2_ molecules and the inherent defects of non-homogeneous reaction systems, the development and design of efficient nitrogen fixation systems face multiple difficulties and challenges, mainly including the low visible light utilization capacity of photocatalytic materials, easy compounding of photogenerated carriers, challenges in the activation of N_2_ adsorption and competition for hydrogen precipitation during the catalytic reaction. Similarly, electrocatalysis also faces many challenges.

### 3.1. Low Utilization of Light Energy

Tuning the bandgap of semiconductor photocatalysts to accept visible light excitation and meet the reduction potential for converting nitrogen to ammonia was necessary for photocatalytic reactions. A narrow band gap was beneficial to improve the efficiency of sunlight utilization, while a too-narrow band gap reduced the energy level of photogenerated electrons, resulting in an insufficient reduction capacity to activate the reactants. For photocatalytic nitrogen fixation, the catalyst CB position must be above the nitrogen reduction overpotential (−0.092 V vs. NHE); the ideal band gap value was about 2.0 eV, corresponding to a light absorption range up to 620 nm. However, the band gap of semiconductor materials (Figure 3) [22] was often beyond the ideal range. For example, conventional TiO_2_ had a band gap value of about 3.2 eV, making it difficult to activate in visible light to generate photogenerated electrons.

Quantum efficiency was an important parameter describing photoelectric conversion in photocatalysis technology. It refers to the ratio of the average number of photoelectrons produced in a specific wavelength per unit of time to the number of incident photons. The quantum efficiency of the catalysts developed in recent years was usually lower than 5%, which is still a low level [23,24,25,26].

### 3.2. Low Separation Rate of Photogenerated Carriers

According to the principle of photocatalytic nitrogen fixation, the nitrogen reduction reaction occurs on the premise that photogenerated electrons are separated from holes and migrate to the catalyst surface. While the migration speed of photogenerated carriers is slow (10^−8^~10^−3^ s) and the compounding process was relatively fast (10^−9^ s), the significant speed difference leads to the photogenerated electrons and holes being extremely easy to compound. It reduces the catalytic reaction efficiency [27]. Therefore, promoting the separation and transfer of photogenerated electron–hole pairs and suppressing photogenerated carrier complexation in photocatalytic reactions is difficult.

How to suppress the migration of photogenerated carriers and reduce the complexation of electron–hole pairs is the focus of the current research on photocatalysis [28,29].

### 3.3. N_2_ Adsorption and Activation Difficulties

N_2_, as an inert molecule, has a robust N≡N bond energy (941 kJ·mol^−1^) and a dissociation energy of up to 410 kJ·mol^−1^ for the first chemical bond, which hinders its cleavage and hydrogenation. According to the front-line orbital theory, the energy gap between the HOMO (σ_g_2p bonding orbital) and LUMO (πg*2p antibonding orbital) of the N_2_ molecule is 10.82 eV, which inhibits the injection process of electrons into the antibonding orbital and hinders the activation of the N_2_ molecule [30]. In the multiphase catalytic process, reactant adsorption on the catalyst surface was a prerequisite for the reaction to occur, and the specific surface area of the catalyst affects the contact between reactants and the exposure of active sites. The formation of chemical bonds between the surface active sites and N_2_ in cooperation weakens the strength of N≡N bonds and promotes the activation of N_2_ molecules. It was the rate-determining step of the photocatalytic nitrogen reduction reaction.

The reduction reaction of nitrogen in the presence of a photocatalyst requires three steps: (1) dissolution of nitrogen in water; (2) diffusion of dissolved nitrogen in water to the liquid surface film of the catalyst; and (3) activation of nitrogen by adsorption on the active sites on the catalyst surface. However, the solubility of nitrogen in the aqueous phase reaction system at room temperature and pressure is extremely low (~1 mmol·L^−1^), and the expansion coefficient is small (10^−5^ cm^2^·s^−1^), resulting in the rate of ammonia synthesis being limited by the process of nitrogen dissolution and diffusion in the reaction system [31,32].

### 3.4. Hydrogen Analysis Competition

Another challenge in photocatalytic nitrogen reduction was the competition with the hydrogen precipitation reaction (HER). Since HER involves only two-electron transfer processes (Equation (1b)), the NRR half-reaction requires a charge transfer process of at least six electrons (Equation (1h)). HER has a kinetic advantage. Second, the HER potential was lower than NRR regardless of acidic or basic conditions, implying that a hydrogen precipitation process inevitably accompanied the NRR reaction. The H_2_ generation leads to a decrease in nitrogen fixation efficiency and a decrease in the selectivity of the photocatalytic reaction [33].

### 3.5. Other Problems in NRR

The electrocatalysis research mainly includes the following aspects: (1) Combining with theoretical calculations further improves the calculation method and model of NRR to provide theoretical guidance for the design of catalysts. (2) The occurrence of HER side reaction reduces the efficiency of the electrocatalytic synthesis of NH_3_. (3) In the process of electrocatalysis, the structure of the intermediates produced by the reaction was complex, and the production time was short. It was difficult to characterize and capture these intermediates by the existing experimental methods.

Research on the electrocatalytic reduction in nitrogen is vital for developing an ecological environment and new energy. The progress in the field of NRR has confirmed that it is possible to reduce N_2_ to NH_3_ using electrocatalysis at a normal temperature and pressure. However, the efficiency of electrocatalysis is low; it is still in the laboratory research stage.

## 4. Modification Strategies for the Photo(electro)catalysts

A photo(electro)catalyst is critical in determining the catalytic reduction reaction. The research focus of nitrogen fixation technology was the construction of efficient photo(electro)catalysts to enhance the reaction activity. The typical photocatalysis process mainly includes three key steps: photoexcited charge generation, photogenerated charge transfer to the catalyst surface and participation in the redox reaction on the surface. Each step is a necessary condition for a catalytic reaction, so the fundamental way to enhance the performance of the catalyst is to improve the efficiency of each step. Various photo(electro)catalyst modification strategies have been developed, such as morphology modulation, heterostructure construction, vacancy introduction, element doping and cocatalyst addition.

### 4.1. Morphology Modulation

Generally, morphological changes could affect the nanocatalyst materials’ physical and chemical properties. Factors such as structure, dimensionality, morphology and size of inorganic nanomaterials directly impact their functional properties, thus affecting their catalytic activity and selectivity [34,35]. Nanomaterials have unique optical, electrical, magnetic, catalytic and mechanical properties closely related to crystal morphology and morphology. Adjusting physical morphology has become an effective method to enhance the photo(electro)catalytic performance by adjusting the size or structure of semiconductors. Therefore, the preparation of controllable particle size and shape semiconductor nanomaterials and the study of their structure and morphology effects on photo(electro)catalytic properties are of great significance for researching and developing new efficient catalytic materials.

#### 4.1.1. Photocatalyst

There was a constitutive relationship between nanomaterials’ size, morphological structure, and properties. Particle size affects the band gap energy, light absorption capacity, and average free range of photogenerated charges, resulting in size quantization, structure-dependent crystallization and the structure-induced multiple reflection effect. According to the order from simple to complex, the structure types of semiconductor materials can be divided into 0/2D, 1D, 3D and multi-dimensional materials [36]. The 0/2D material refers to solid film, film or powder, which has the advantages of easy preparation and convenient utilization. At the same time, the plane shape on the nanometer scale will change the position of its band gap, conduction band and valence band. One-dimensional materials, such as nanoparticles, nanowires and nanorods. It could be a semiconductor with one-dimensional ordered geometry. Such materials often have a high specific surface area, which can shorten the charge transfer distance and provide rich active centers and a large light absorption area for redox reactions. Multi-dimensional materials such as 3D have more complex structures, which further increases the surface area of semiconductor materials, and shows structure-induced characteristics such as multiple reflections of incident light.

For example, in recent years, 2D semiconductors have aroused great interest in photocatalysis due to their remarkable advantages [37]. One is that some 2D semiconductors have a variable band gap through size adjustment, thus improving the light absorption ability. At the same time, the sharp decrease in semiconductor thickness shortens the migration distance of photogenic charge from the bulk phase to the surface phase, which is advantageous to the separation and transportation of photogenerated charge. In addition, for the ultrathin properties of 2D materials, the abundant reaction sites generated by a high specific surface area will also be favorable for photocatalytic reactions. Gao et al. [38] prepared AgCl/δ-Bi_2_O_3_ nanosheets with a thickness of about 2.7 nm by hydrothermal precipitation. The structural properties of the catalysts, including morphology, crystallinity, optical properties and energy band structure, were analyzed by SEM, TEM and AFM tests. The characterization results show that the ultrathin two-dimensional nanomaterials have a larger specific surface area and two-dimensional anisotropy, exposing more surface unsaturated paired atoms, increasing the active surface centers and shortening the charge migration distance to achieve effective separation of electron–hole pairs. Meanwhile, surface atoms tend to escape from the lattice during the size reduction in nanomaterials, inducing surface defects. Feng et al. [39] successfully prepared surface oxygen vacancy-modified Bi_2_O_2_CO_3_ (i.e., BOC/OV) by continuously compressing the size of nanomaterials at room temperature. The size of the individual sheets has been reduced to 10 × 10 nm, the crystalline periodic boundary conditions are disrupted, and the atomic density near the surface layer of the nanoparticles is diminished. Under the synergistic effect of surface vacancies and the small size effect, the adsorption activation of BOC/OV on N_2_ is enhanced, the complexation rate of photogenerated carriers is reduced and the photocatalytic reduction performance is enhanced to 10 times the original BOC yield. In addition, the hollow structure can slow the escape of nitrogen and reaction intermediates and prolong the nitrogen residence time due to cavities while shortening the electron migration distance and inhibiting photogenerated carrier complexation. Vu et al. [40] calcined MIL-68-In(Ru) precursors in the air to obtain Ru-doped In_2_O_3_ materials. The nanoparticles are relatively uniformly distributed and self-assembled into a hollow peanut structure, which facilitates the separation and transport of photogenerated electrons and holes and improves the light energy utilization through multiple reflection and scattering processes of light. For analysis and comparison, catalysts with specific morphological structures from the above and recent studies are compiled in Table 2, and their morphological modulation methods, photoreaction conditions and ammonia production efficiency are summarized.

#### 4.1.2. Electrocatalyst

For electrocatalysis, adjusting the morphology of the catalyst is also a powerful strategy to improve the performance of NRR. Li et al. reported that porous palladium ruthenium nanosheets (PdRu NS-NF) were grown in situ on Ni foam with Ni foam as a reducing agent and substrate [48]. These ultrathin nanosheets have a high density of unsaturated atoms and a high specific surface area, providing rich active sites and available channels for charge transport. The ammonia yield and Faraday efficiency (FE) of PdRu NS-NF without adhesive are 20.46μg·h^−1^cm^−1^ and 2.11%, respectively, showing excellent activity and stability in the electrocatalytic reduction in nitrogen.

In addition, low coordination sites can be formed by adjusting the size of the electrocatalyst, which affects the binding strength of the reaction intermediates, thus optimizing the activity and selectivity of the electrocatalyst. Shi et al. successfully prepared sub-nanometer (≈0.5 nm) Au clusters on TiO_2_ substrate and formed stable Au-O-Ti bonds through lattice oxygen [49]. Three kinds of Au/TiO_2_ catalysts with different metal sizes were prepared simultaneously for comparison. The experimental results show that due to the high surface energy of small metal clusters, strong chemical bonds were always formed between the coordination lattice oxygen of atomic metal and oxide to maintain the stability of the metal. Therefore, the sub-nanometer Au clusters anchored on TiO_2_ have stability under long-term electrolysis conditions, significantly improving the ammonia yield of electrocatalytic NRR (21.4 μg·h^−1^ mg^−1^_cat_) and FE (8.11%). It is much better than the best effect of N_2_ fixation under environmental conditions and even comparable to the yield activation energy at high temperature and/or high pressure. Furthermore, the active site of the catalyst could be improved effectively by adjusting the crystalline surface of the catalyst through good atomic arrangement and coordination. However, there is little research on applying crystal surface modulation to electroreducing nitrogen reactions, so further research in this area can be conducted.

### 4.2. Heterostructure Construction

Heterojunctions refer to the interfacial region formed after the contact between two different semiconductors; the construction of heterojunctions enhances the light absorption ability of catalysts and improves the quantum efficiency with the help of the window effect, and can promote the separation of photogenerated electrons and holes by using the interfacial impact [50,51]. Generally, the heterojunctions formed between different semiconductors were classified into three types of conventional heterojunctions (including type-I, type-II and type-III), p-n heterojunctions and Z-scheme heterojunctions (including conventional Z-scheme, all-solid Z-scheme and direct Z-scheme).

Fan et al. [52] used a precipitation method to grow In(OH)_3_ nanoparticles on g–C_3_N_4_ nanosheets to synthesize 0D/2D type-I heterojunctions (Figure 4a). Although In(OH)_3_ in the heterojunction cannot produce photogenerated electrons in visible light, its defect energy level was close to the CB of g–C_3_N_4_, which enables it to trap excited-state electrons of g–C_3_N_4_, improving the separation efficiency of electron–hole pairs, extend the lifetime of photogenerated carriers, and obtain excellent nitrogen fixation activity and stability. Mou et al. [53] used terephthalic acid as a molecular probe. The synthesized g–C_3_N_4_/ZrO_2_ was verified to be a type-II heterojunction by capturing the hydroxyl group by fluorescence (Figure 4b). Among them, g–C_3_N_4_ has excellent light absorption ability and can be used for light trapping. The photogenerated electrons migrate from the CB of g–C_3_N_4_ to the CB of ZrO_2_ because of the type-II heterojunction. It promotes the reverse movement of photogenerated electrons and holes, reducing the complexation rate of photogenerated carriers. P-n type has a similar structure to the type-II heterojunction, and in p-n-type photocatalysts, the CB and VB energy levels of p-type semiconductors are usually higher than those of n-type semiconductors. Due to the internal electric field and energy band, the p-n heterojunction has a faster charge separation rate than the type-II heterojunction due to the synergistic effect of the internal electric field and energy band alignment [54]. The NiS/KNbO_3_ prepared by Zhang’s research group [55] has a strong N_2_ reduction ability under simulated solar irradiation. Since the Fermi energy level of KNbO_3_ was higher than that of NiS, electrons spontaneously migrate from KNbO_3_ to NiS, which causes the energy band potential of NiS to increase and the composite material to constitute a p-n heterojunction and generate an internal electric field. Driven by the energy band structure and the built-in electric field, electrons in NiS were transferred to the CB of KNbO_3_, while holes in KNbO_3_ were transferred to the VB of NiS, and the photogenerated electrons and holes achieved efficient spatial separation (Figure 4c). Thus, NiS/KNbO_3_ exhibited good photocatalytic efficiency in N_2_ reduction, which was 1.9 and 6.8 times higher than that of KNbO_3_ and NiS, respectively. Inspired by artificial photosynthesis, the concept of Z-scheme heterojunction was further proposed to improve the photocatalytic system’s redox potential. Duan et al. [56] synthesized three-dimensional Cu_2_O/MoS_2_/ZnO nanocomposites with copper mesh as a substrate for photocatalytic N_2_ reduction experiments under visible light. Excellent interfacial contact achieved a high ammonia yield, which was 4.24 and 2.58 times higher than ZnO and Cu_2_O. Characterization tests showed that the photogenerated charges in the composite showed a double Z-scheme transport path, the weak oxidation holes and weak reduction electrons in the charge transfer process were directly quenched and the photogenerated carrier separation efficiency and catalyst reduction capacity were significantly enhanced. At the same time, a broad spectral response and high stability were achieved (Figure 4d).

At present, heterojunction electrocatalysts are mainly concentrated on type-II catalysts. Hu et al. [57] have proven that compared with g–C_3_N_4_ or ternary metal sulfide, the construction of g–C_3_N_4_ with ternary metal sulfide could significantly improve the nitrogen fixation efficiency. When the mass fraction of ZnMoCdS was 80%, the heterojunction system had the highest nitrogen fixation capacity. A large number of sulfur vacancies acted as active centers and significantly improved electrocatalytic activity.

### 4.3. Introduction of Vacancies

Defect engineering was an effective strategy to modulate materials’ electronic and chemical structure. Defects were mainly classified into four categories based on size: point defects, line defects, surface defects and bulk defects. Among them, point defects were a type of defect that deviates from the crystal structure or normal arrangement in the lattice nodes or adjacent microscopic regions, which were mainly classified into oxygen vacancies (OV), sulfur vacancies (SV) and nitrogen vacancies (NV) according to elemental categories [58]. Whether in photocatalysts or electrocatalysts, how to introduce oxygen vacancies (OVs)—which are anionic vacancies with a low free-energy formation and are usually easily formed in transition metal oxides—is the topic that is most studied. The created oxygen vacancies can be used as active sites for adsorption reactants and reaction intermediates, lowering the activation energy barrier and promoting electrochemical reactions. In addition, the introduction of vacancies can affect the electronic structure of catalysts and enhance charge. In addition, the introduction of vacancies can affect the electronic structure of the catalyst, enhance charge transfer and expose more metal sites with unsaturated coordination.

#### 4.3.1. Photocatalyst

The modification effect of introducing vacancies in materials was generally reflected in three aspects: (1) the generated vacancies could act as trapping centers for photogenerated electrons or holes and inhibit photogenerated charge complexes; (2) they act as active centers and promote the adsorption and activation of reactant molecules; and (3) they adjust the band gap energy of catalysts and enhance the light absorption capacity. Cao et al. [59] used metal doping such as Ni or Mo to induce distortion in the lattice of CdS, resulting in vacancy defects on the material surface, and performed photocatalytic nitrogen reduction tests. There was a positive correlation between the sulfur vacancy concentration and ammonia generation rate within a certain reaction time, where Mo_0.1_Ni_0.1_Cd_0.8_S contained the highest sulfur vacancy concentration, and its ammonia yield was correspondingly the highest (Figure 5a). Combined with the analysis of ICP, TPD, and nitrogen fixation reaction results, the surface sulfur vacancies could not only serve as active sites for the adsorption and activation of N_2_ molecules, but could also promote the interfacial charge transfer from the catalyst to the N_2_ molecules, thus significantly improving the nitrogen photo fixation capacity.

In addition, the ability of nitrogen vacancies to selectively use nitrogen as an adsorbed species due to matching the shape and size of nitrogen atoms implies that the photocatalytic reaction can directly utilize air as a nitrogen source and reduce the production cost. Dong et al. [60] first reported that nitrogen vacancies enable g–C_3_N_4_ to selectively adsorb and reduce N_2_ in gas mixtures without interference from other gases. By the density flooding theory (DFT) calculations, N_2_ molecules tend to adsorb on nitrogen vacancies and form σ-bonds with two adjacent C atoms. The coupling mode of N_2_ molecules changes as the nitrogen–nitrogen bond length is extended from 1.117 Å to 1.214 Å, thus demonstrating that the nitrogen vacancies activate N_2_. In addition to the work on nitrogen vacancies, a recent research team introduced carbon vacancies on the surface of g–C_3_N_4_ (SCNNSs) catalysts using sulfur doping and synthesized nanosheets with a graded pore structure and high specific surface area [61]. DFT simulated the adsorption configuration of N_2_ on SCNNSs–550, and the optimization results showed that the carbon vacancies more easily adsorbed N_2_ molecules in SCNNSs–550 with a calculated adsorption energy of −0.665 eV, which is 1.99 times higher than that of bulk SCN. The charge density difference of the molecules (Figure 5b) showed that the charge separation efficiency was increased by transferring photogenerated electrons from the carbon vacancies in SCNNSs–550 to N_2_ molecules. In contrast, N_2_ molecules were activated to form high-energy intermediates (–N_2_H or HN = NH) to promote the photocatalytic nitrogen fixation reaction.

#### 4.3.2. Electrocatalyst

Based on DFT theoretical calculations, Yao et al. [62] proposed a strategy to optimize the NRR performance of BiVO_4_ catalysts by regulating the oxygen vacancy concentration. They successfully prepared efficient BiVO_4_ catalysts containing different OVs concentrations by a simple hydrothermal method with reasonable pH adjustment. The highest concentration of oxygen vacancies was obtained at pH = 7, and the best nitrogen reduction performance was achieved. The ammonia synthesis rate was 8.6 μg·h^−1^·mg^−1^_cat_ at −0.5 V vs. RHE potential with a Faraday efficiency of 10.04%. It was confirmed that oxygen vacancies play an essential role in nitrogen reduction. The introduction of oxygen vacancies in the electrocatalyst can enhance the adsorption of N_2_ adsorption and optimize NRR activity. Researchers could take advantage of the present experience in the construction of oxygen vacancies that has been reported in order to develop more efficient oxygen-containing vacancy catalysts, thus promoting the development of this new field of NRR.

Oxygen and sulfur are located in the same main group and have similar chemical properties. The researchers speculated that the sulfur vacancies (SVs), similar to OVs, could influence the catalysts’ energy band structure and electronic structure and act as N_2_ adsorption sites to induce N_2_ activation, thus affecting their electrocatalytic NRR performance [63].

Similar to photocatalysis, the introduction of nitrogen heteroatoms in electrocatalysis is required to create a net positive charge on the adjacent carbon atoms, thus facilitating the adsorption and activation of N_2_ on the catalyst. The experimental results showed that the NH_3_ yield was 8.09 μg·mgcat.^−1^·h^−1^ with a Faraday efficiency of 11.59%, which was about 10 times higher than that of the catalyst without nitrogen vacancies [64].

Due to the stable structure of N_2_ itself, it is weakly combined with heterogeneous catalysts. It hinders the development of photocatalytic nitrogen fixation, which requires changing the structure of the catalyst to activate the N atom. Vacancies (oxygen vacancies, sulfur vacancies, nitrogen vacancies) could effectively modulate the catalyst’s electronic structure and band gap and optimize the charge distribution of the catalyst. In addition, vacancies could adsorb and activate inert N_2_ molecules, lowering the reaction barriers and improving photocatalytic efficiency [65,66,67,68,69,70,71,72].

### 4.4. Cocatalyst Addition

Cocatalyst was a conductor (or semiconductor) coupled with a semiconductor to improve the overall catalytic activity, which improves the performance of photocatalysts in three main ways: (1) reducing the activation energy or overpotential of redox half-reactions; (2) promoting electron–hole pair separation and inhibiting photogenerated charge complexes; (3) inhibiting photocorrosion and improving the stability of photocatalytic materials. The two commonly used cocatalysts in the nitrogen fixation reaction system were reduced and plasma cocatalysts [73,74,75,76,77].

MXene has excellent electrical, optical and thermodynamic properties. It was expected to replace noble metal cocatalysts as a new, reduced cocatalyst. Liao et al. [78] used 2D–layered Ti_3_C_2_ MXenes as a cocatalyst for P25, and the ammonia yield of the optimized 6% Ti_3_C_2_ MXenes–P25 sample under full-spectrum irradiation was 10.74 μmol·g^−1^·h^−1^, which was five times higher than that of pure P25. The experimental tests and DFT results showed that Ti_3_C_2_ MXenes played an essential role in photogenerated electron–hole separation and N_2_ adsorption activation by storing electrons from the excited-state P25 (Figure 6a). Secondly, Ti_3_C_2_ MXenes can facilitate the reaction between adsorbed N_2_ and the stored photoelectrons and are proven to have an effective cocatalyst. Carbon quantum dots have attracted much attention as a new class of carbon nanomaterials because of their low cost, low toxicity, chemical inertness, superior water solubility, easy functionalization and simple synthesis routes. Wang et al. [79] developed a composite system of biological nitrogen–fixing bacteria modified by carbon quantum dots (CDs). CDs have the properties of electron donors and acceptors. Their binding to nitrogen-fixing enzymes changed the α–helix of nitrogen-fixing enzymes’ structure, accelerating the electron transfer in the catalytic process and significantly increasing the nitrogen-fixing enzyme activity (Figure 6b). The excellent light capture ability and unique photogenerated electron transfer ability of carbon quantum dots proved to be a potentially reduced cocatalyst with promising applications in the field of photocatalytic nitrogen fixation.

Gold nanoparticles (Au NP) produce a localized surface plasmon (LSPR) effect when used as a cocatalyst. Under visible light irradiation, gold nanoparticles deposited on the surface of the semiconductor induce the production of high–energy electrons (hot electrons). These electrons can overcome the Schottky barrier and are irreversibly injected into the semiconductor’s conduction band. The photogenerated electrons and holes are spatially separated in the semiconductor and metal nanoparticles, thus inhibiting photogenerated charge complexation. Nazemi and EI-Sayed [80] used hollow Au-Ag_2_O nanocages to convert N_2_ to NH_3_ in a pure water system with an average ammonia yield of 28.2 mg·m^−2^·h^−1^ and an apparent quantum efficiency of 1.2% under simulated sunlight irradiation (the light source was monochromatic light at 685 nm). Au generates not only hot holes and electrons through LSPR excitation, which provides the active center of water oxidation and nitrogen reduction, but also collects photogenerated electrons from Ag_2_O, inhibits the electron–hole complexation, improves the charge transfer efficiency of NRR and significantly enhances the photocatalytic NRR activity (Figure 6c). In addition to Au nanoparticles, other metal-based materials such as Ag, Cu and Ru can also achieve the LSPR effect. Mao et al. [81] developed a photothermal catalytic system using multifunctional Ru nanoparticles as plasma-assisted catalysts and adsorption activation sites for N_2_ molecules. TiO_2-x_H_x_ has abundant OVs that increase the electron density around Ru nanoparticles. Based on the LSPR-induced binding energy, H donation and feedback between the TiO_2-x_H_x_ surface and the Ru catalyst could be achieved. Therefore, ammonia could be synthesized on the Ru surface by an acceptor mechanism (Figure 6d, Mechanism I) and generated by a reaction between Ru-activated N and Hinc on the TiO_2-x_H_x_ surface (Figure 6d, Mechanism II). This reaction mechanism prevents the poisoning of the Ru surface by H during thermal synthesis and ensures the high efficiency and stability of nitrogen fixation [26,82,83,84,85,86].

Oshikiri et al. [87] doped the strontium titanate photoelectrode with Au NPs deposited on the surface by niobium immersed in the anode with ethanol as a sacrificial donor. A Ru-based cocatalyst was used to modify Nb–SrTiO_3_ with an N_2_–saturated aqueous HCl solution. The spacer of this photoelectrochemical system effectively prevented the further oxidation of ammonia to nitrate during the catalytic process. In addition, the potential chemical difference was generated by the different pH values of the cathode and anode chambers, thus accelerating the rate of NH_3_ synthesis.

Liu et al. [88] synthesized Co element-doped sulfur vacancy-rich MoS by introducing sulfur vacancies at the MoS basal position and using doped Co elements to replace Mo in the lattice. The catalyst showed excellent NRR activity. The best Faraday efficiency was 10%, with an ammonia yield of 0.63 mmol·h^−1^·g^−1^. It was attributed to the fact that the Co element doping accelerated the adsorption and dissociation of N_2_ on the defective MoS, promoting the NRR process.

### 4.5. Computational Modeling

The current computational modeling of photocatalytic and electrocatalytic reactions was mainly based on the density function theory (DFT) calculations. Gao et al. [89] used DFT calculations to load single atoms Pd and Pt onto g–C_3_N_4_, respectively. It significantly reduced the reaction potential and enhanced the photocatalytic reduction in CO_2_ to HCOOH and CH_4_. Chu et al. [90] delved into the role of Mo doping in MnO_2_ catalysts and further evaluated the rationality of the distal binding mechanism by DFT calculations. Concerning the reported theoretical study and the fact that N_2_H_4_ was not found in the reaction products, a possible distal binding mechanism of Mo_3_/Fe_2_@ZC–cm photocatalyst during hydrogenation was proposed. In the nitrogen reduction reaction, the photocatalyst provides abundant photogenerated electrons, water provides protons (H^+^), and the metal sites near the oxygen vacancies could act as active centers to adsorb and activate N_2_ molecules but also facilitate the charge transfer from the photocatalyst to the N_2_ molecules. In this environment, the Mo/Fe factor adsorption activates the distal N atom in the N_2_ molecule to bind to H^+^, weakening the stable N≡N triple bond. The NH_3_ molecule was formed and successfully desorbed in the subsequent proton-coupled electron transfer process. The N atom of the other one continued to bind to H^+^ until another NH_3_ was formed and released. At this point, the reaction process of N_2_ reduction to NH_3_ was completed, and the next cycle proceeded [91,92,93,94,95].

## 5. Conclusions and Prospects

Photo(electro)catalysis uses sunlight and electricity to reduce nitrogen to ammonia through catalysts. It provides a promising development path for achieving green and sustainable ammonia production. This paper reviews and discusses the research progress on photo(electro)catalytic nitrogen reduction. After years of efforts, photo(electro)catalytic nitrogen fixation technology has made significant progress but has not yet reached the requirement of industrial production. To promote the practical application of the technology for nitrogen fixation, future research work prospects and the following points are proposed in this field.

Formulate a unified evaluation standard for nitrogen fixation systems. Various factors, including reaction equipment, reaction temperature and pressure, light source and intensity, and product detection methods, influenced the stability of photo(electro)catalytic efficiency. In addition, the evaluation standard of the photocatalytic NRR reaction is usually based on the absolute yield and evolution rate of ammonia production (μmol·g_cat_^−1^ h^−1^ or μmol·h^−1^). With the innovation and development of the photocatalytic reaction system, reactors’ design types have been enriched [92]. The application form of photocatalyst is no longer a single suspension type, and the supported catalyst that is convenient for recovery and utilization has also begun to receive attention [93]. How to make a uniform and fair comparison of the performance of these different types of photocatalysis systems has become a thorny problem. In addition to specifying various parameters in detail in the report, the number of substances that mainly play a catalytic role and the corresponding active sites are discussed. Therefore, a reliable and strict evaluation standard for the photo(electro)catalytic nitrogen fixation must be established to ensure the reliability and comparability of the experimental data.Because of the low reaction efficiency caused by the poor solubility of the N_2_ molecule, it is considered that the nitrogen source used in the photo(electro)catalytic reaction can be replaced. Nitrogen-based compounds such as nitrate, nitrite and nitrogen oxide are readily soluble in water. Therefore, the problem of N≡N cracking and activation could be avoided, and the hydrogen evolution reaction could be inhibited. Similarly, water vapor can also be used as the proton source. Simplifying the traditional gas–liquid–solid three-phase reaction into a gas–solid two-phase reaction is a potential method to improve the efficiency of the NRR reaction.Recently, a first-principle calculation combined with kinetic analysis has been widely used to predict the reaction potential barrier of the rate-determining step. However, the precise reaction kinetics theory has not been determined and is still developing. It is necessary to conduct more in-depth thermodynamic and kinetic studies to understand the catalytic performance of ammonia synthesis more practically. For example, the photo(electro)catalytic nitrogen reduction process was studied at the molecular or atomic level by combining experiments and theoretical calculations. Although some of the studies combine theory and experiment, most theoretical calculations focus on the free–energy change in the active intermediate—the energy barrier of the rate-limiting step. These calculations of the adsorption energy of N_2_ are assumed to be performed under vacuum conditions, ignoring the influence of the determinants of the electrochemical system (e.g., temperature, pH, mass transfer rate, proton supply, N_2_ solubility), which are different from the actual experimental conditions. Given these problems, in-situ experimental techniques can be used to capture and identify reaction intermediates and monitor the microscopic changes of catalysts to assist theoretical research. Therefore, it is necessary to further improve the calculation method and model of NRR on the surface of non-homogeneous catalysts to combine theoretical calculations and experiments and to provide further guidance for designing electrocatalyst structures.Currently, the catalytic activity and selectivity of catalysts for ammonia synthesis in aqueous solutions are extremely low. One of the main factors is the presence of competition from side reactions. Therefore, the catalytic activity and selectivity can be significantly improved by suppressing the occurrence of side reactions. Specifically, optimizing the size and morphology of catalysts can generate favorable coordination sites, influence the binding strength of reactants or key intermediates on the catalyst surface and construct defect engineering of catalysts. It was necessary to capture photoelectrons and sub-stable electrons through vacancies and transfer them into the antibonding orbitals of adsorbed N_2_ to promote the breakage of N≡N bonds. The construction of stress engineering to regulate the atomic surface spacing and bond length also can change the catalyst’s electronic structure, thus facilitating the nitrogen reduction reaction. In addition, while exploring effective catalysts and constructing excellent photocatalytic systems for future research, we should also pay attention to the efficiency and stability issues.

## Figures and Tables

**Figure 1 molecules-28-02666-f001:**
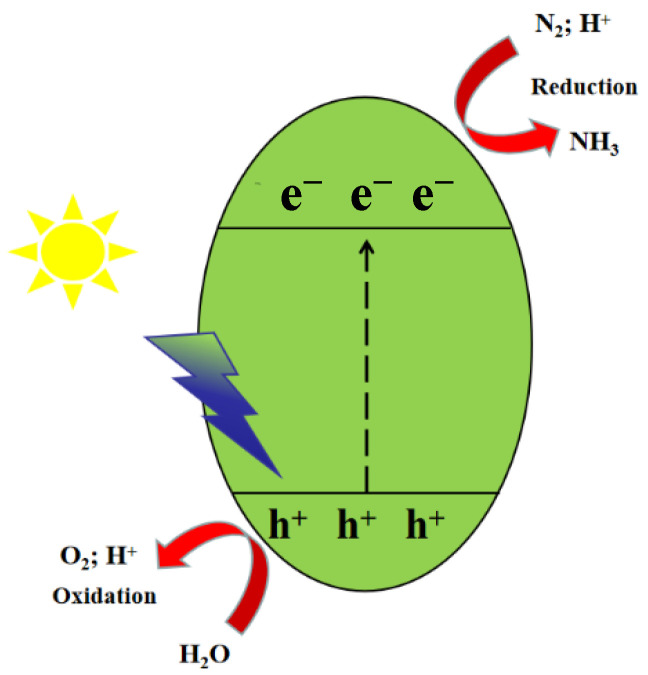
Schematic illustration of the photocatalytic process of semiconductors.

**Figure 2 molecules-28-02666-f002:**
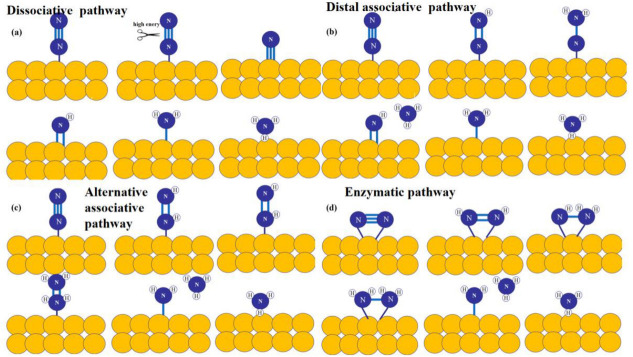
The mechanism of nitrogen fixation (**a**) dissociative, (**b**) distal associative, (**c**) alternative associative, (**d**) enzymatic.

**Figure 3 molecules-28-02666-f003:**
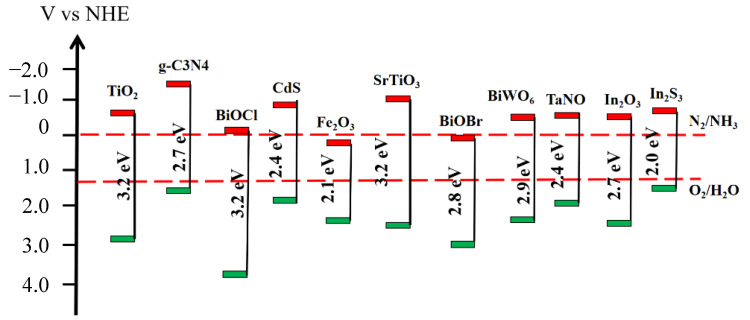
Band–edge positions for conventional semiconductors (vs. NHE at pH = 0).

**Figure 4 molecules-28-02666-f004:**
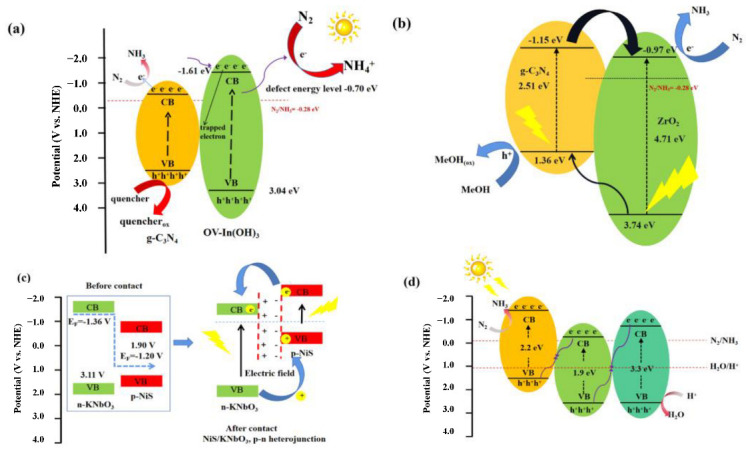
Schematic illustrations of photocatalytic N_2_ fixation over (**a**) OV-In(OH)_3_/g–C_3_N_4_, (**b**) g–C_3_N_4_/ZrO_2_, (**c**) NiS/KNbO_3_, (**d**) Cu_2_O/MoS_2_/ZnO–cm.

**Figure 5 molecules-28-02666-f005:**
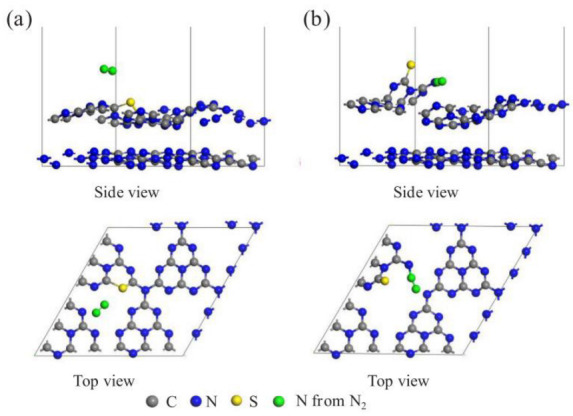
The optimized geometry of the optimal N_2_ adsorption models on (**a**) bulk SCN and (**b**) SCNNSs–550. Reprinted with permission from Ref. [59]. Copyright 2018, Elsevier.

**Figure 6 molecules-28-02666-f006:**
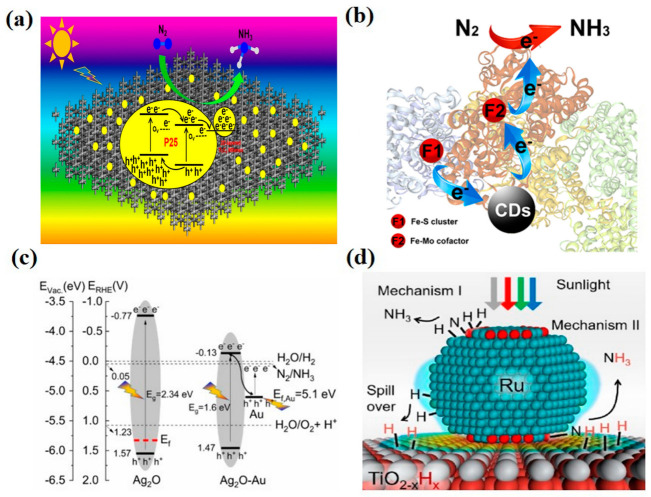
(**a**) Schematic illustration for the promotion effect of Ti_3_C_2_ MXenes on P25 for photocatalytic NRR, reprinted with permission from Ref. [78]. Copyright 2020, Elsevier. (**b**) Schematic diagram of electron transfer pathways for nitrogenase/CD hybrids, reprinted with permission from Ref. [22]. Copyright 2019, John Wiley and Sons (**c**) Electronic band structure of the Ag_2_O and Ag_2_O-Au photocatalyst for the nitrogen reduction reaction, reprinted with permission from Ref. [80]. Copyright 2019, Elsevier (**d**) Schematic illustration of solar thermal ammonia synthesis on Ru/TiO_2-x_H_x_. Reprinted with permission from Ref. [81]. Copyright 2018, Elsevier.

**Table 1 molecules-28-02666-t001:** Nitrogen hydrogenation reaction and reduction potentials (vs. NHE at pH = 0).

Reaction	Reduction Potential (V)	Equation
H_2_O → 1/2O_2_ + 2H^+^ + 2*e*^−^	0.81	1a
2H^+^ + 2*e*^−^ → H_2_	−0.42	1b
N_2_ + *e*^−^ → N_2_^-^	−4.16	1c
N_2_ + H^+^ + *e*^−^ → N_2_H	−3.2	1d
N_2_ + 2H^+^ + 2*e*^−^ → N_2_H_2_	−1.10	1e
N_2_ + 4H^+^ + 4*e*^−^ → N_2_H_4_	−0.36	1f
N_2_ + 5H^+^ + 4*e*^−^ → N_2_H_5_^+^	−0.23	1g
N_2_ + 6H^+^ + 6*e*^−^ → 2N_2_H_3_	0.55	1h
N_2_ + 8H^+^ + 8*e*^−^ → 2N_2_H_4_	0.27	1i

**Table 2 molecules-28-02666-t002:** Comparison of recent photocatalysts with specific morphologies and nitrogen fixation properties.

Photocatalyst	Morphological Characteristics	Preparation Method	Nitrogen Source	Sacrificial Agent	Light Source	Ammonia Yield	References
Ag/PM-CdS(e)	Nanospheres (diameter of about 14.2 nm)	Hydrothermal–Etching	N_2_	-	λ > 420 nm	0.343 μg·h^−1^·mg^−1^	[41]
AgCl/δ-Bi_2_O_3_	Nanosheets (thickness of about 2.7 nm)	Hydrothermal precipitation method	N_2_	-	λ > 420 nm	606 μmol·h^−1^·g^−1^	[42]
BOC/OV_3_	Micro-nanosheets (<10 × 10 nm)	Room temperature reaction-reduction	N_2_	Na_2_SO_3_	λ > 400 nm	1178 μmol·L^−1^·g^−1^·h^−1^	[43]
Ru-In_2_O_3_	Hollow peanut structure	Air Calcination	N_2_	Methanol	UV-vis	44.5 μmol·g^−1^·h^−1^	[40]
NYF/NV-CNNTs	Nanotubes	Solvothermal method	N_2_	Ethanol	λ~980 nm or > 420 nm	1.72 mmol·L^−1^·g_cat_^−1^ or 5.30 mmol·L^−1^·g_cat_^−1^	[44]
Au/HCNS-NV	Mesoporous hollow spheres	Templating agent calcination-reduction	N_2_	Methanol	λ > 420 nm	783.4 μmol·h^−1^·g_cat_^−1^	[45]
1T’-MoS_2_/CNNC	Nanocages	Hydrothermal method	N_2_	Methanol	UV-vis	9.8 mmol·L^−1^·h^−1^·g^−1^	[46]
N*v*&O*d*-CN	Porous hollow prisms	Low-temperature hydrothermal–calcination	N_2_	Methanol	λ > 420 nm	118.8 mg·L^−1^·h^−1^·g_cat_^−1^	[47]

## Data Availability

Not applicable.

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
