# Peer review of "Advances in Semiconductor-Based Nanocomposite Photo(electro)catalysts for Nitrogen Reduction to Ammonia"

_molecules, 2023, doi:10.3390/molecules28062666_

Round 1

Reviewer 1 Report

The manuscript titled “Advances in Semiconductor-Based Nanocomposite Photocatalysts and Electrocatalysts for Nitrogen Reduction to Ammonia” is an interesting work. The review described the process and mechanism of nitrogen reduction using semiconductor nanomaterials, and points out the main challenges of the photocatalytic nitrogen fixation system. The review article can be proven of certain significance. This work could be considered for publication and prove to be more interesting if the authors made the following changes/modifications.

1.     The background of catalytic nitrogen reduction to ammonia and role of semiconductor materials needs to be further strengthened in the introduction section.

2.     The author can also inculde the influence of material microstructure on the performance mechanism.

3.     The progressive relationship between each work is week, which needs further combing. The innovation points of each work should be well reflected.

4.     Some important work needs to be introduced in the content, e.g. https://www.mdpi.com/2079-4991/11/10/2548);  https://pubs.rsc.org/en/content/articlelanding/2022/VA/D2VA00018K

5.     The conclusion and outlook are not deep enough to guide peers. The problems encountered in the current research, especially the regulatory mechanism, should be analyzed in more detail. This section should be considered carefully and highlighted.

6.     There are some language errors and description errors in the article, which can be modified to make the article more beautiful.

Author Response

Dear Editors and Reviewers,

Thanks for your comments concerning our manuscript entitled “Advances in Semiconductor-Based Nanocomposite Photocatalysts and Electrocatalysts for Nitrogen Reduction to Ammonia” (manuscript ID molecules-2247017). Those comments are all valuable and helpful for revising and improving our paper. We have carefully studied the comments and made corrections based on your comments. Modified portions are highlighted in red in the revised version. As suggested, we improved the manuscripts by improving the writing and adding more data and analyses. The revised manuscript is responded to, point by point.

The primary corrections in the paper and the responses to the reviewer’s comments are as follows.

Thank you and best regards.

Yours sincerely,

Corresponding author: Qian Su suqian93@163.com

The manuscript titled "Advances in Semiconductor-Based Nanocomposite Photocatalysts and Electrocatalysts for Nitrogen Reduction to Ammonia" is an interesting work. The review described the process and mechanism of nitrogen reduction using semiconductor nanomaterials, and points out the main challenges of the photocatalytic nitrogen fixation system. The review article can be proven of certain significance. This work could be considered for publication and prove to be more interesting if the authors made the following changes/modifications.

  1. The background of catalytic nitrogen reduction to ammonia and role of semiconductor materials needs to be further strengthened in the introduction section.

Response: Thanks for the comment. The introduction part was modified to supplement the background of nitrogen reduction to ammonia and the role of semiconductor materials.

“Ammonia is a carbon-free hydrogen storage compound widely used in agriculture, the chemical industry, medicine, energy storage, and other fields [1]. Ammonia Haber-Bosch (HB) synthesis process is undoubtedly one of the most important inventions in modern history and is also the most mature process for the large-scale production of nitrogen-based fertilizer. The annual output of NH3 reaches 160 million tons, of which 80% is used to produce fertilizer to feed more than 70% of the world's population. The HB process uses an iron-doped catalyst to convert N2 and H2 mixed in a 1:3 ratio to ammonia. This process requires reaction conditions of high temperature (400-600 oC) and high pressure (200-400 atm) to activate the highly stable N≡N (bond dissociation energy up to 941 kJ·mol-1) in the N2 molecule. But the HB process consumes about 2% of the world's total energy annually and emits 300 million tons of CO2 greenhouse gas, accounting for about 1.6% of global emissions [2]. Therefore, to alleviate energy consumption and environmental problems, seeking a sustainable and cost-effective ammonia synthesis strategy is necessary. NH3 synthesis strategies such as biocatalysis, electrocatalysis, and photocatalysis have been studied on a laboratory scale. Among them, photo(electro)catalytic nitrogen reduction synthetic ammonia technology, driven by inexhaustible solar energy, reduces nitrogen to ammonia with high energy density and easy storage and transportation. In addition, the nitrogen reduction process can also be carried out under environmental conditions while achieving zero carbon emission, which is harmless to the environment. It is considered a potential alternative to the industrial HB process to generate NH3, which has aroused the keen attention of society.

Although the Gibbs free energy of ammonia synthesis is negative, it cannot spontaneously react at room temperature and environmental pressure due to the stability and chemical inertness of N2. It is an arduous task to convert N2 into NH3 directly. However, the heterogeneous reaction efficiency in an aqueous solution is low due to the weak adsorption and the activation difficulty of N2 molecules on the catalyst surface, the participation of high-energy intermediates, and the complex multi-electron transfer reaction pathway. In addition, the photocatalytic reduction reaction is affected by the easy recombination of photogenerated electron holes and the weak reduction ability of photogenerated electrons. The reaction efficiency of electrocatalysis is limited by high overpotential, low current density, low selectivity, and competition for hydrogen evolution reactions. Therefore, developing highly active and stable catalysts is vital in converting N2 to NH3.”

“Photo(electro)catalysis has developed rapidly in the past decades, and many different types of photo(electro)catalysts have been proposed, including noble metals, metal complexes, organic molecules, ions, and semiconductors. Since semiconductors offer significant advantages in manufacturing cost, material toxicity and stability compared to other situations, they show attractive promise in photocatalysis research and have long been studied and tested.”

  1. The author can also inculde the influence of material microstructureon the performance mechanism.

Response: Thanks for the comment. In section 4, the relevant content of the influence of material microstructure on the performance mechanism was supplemented.

“The typical photocatalysis process mainly includes three key steps: photoexcited charge generation, photogenerated charge transfer to the catalyst surface, and participation in the redox reaction on the surface. Each step is a necessary condition for catalytic reaction, so the fundamental way to enhance the performance of the catalyst is to improve the efficiency of each step. Various photo(electro)catalyst modification strategies have been developed, such as morphology modulation, heterostructure construction, vacancy introduction, element doping, and cocatalyst addition.”

  1. The progressive relationship between each work is weak, which needs further combing. The innovation points of each work should be well reflected.

Response: Thanks for the comment. The content of the article was reorganized, and the relevant content was deleted or supplemented in each section to highlight the innovation.

  1. Some important work needs to be introduced in the content, e.g. https://www.mdpi.com/2079-4991/11/10/2548

https://pubs.rsc.org/en/content/articlelanding/2022/VA/D2VA00018K

Response: Thanks for the comment. These two articles are insightful on the effect of semiconductor size on catalytic performance and have been cited in the paper.

  1. The conclusion and outlook are not deep enough to guide peers. The problems encountered in the current research, especially the regulatory mechanism, should be analyzed in more detail. This section should be considered carefully and highlighted.

Response: Thanks for the comment. It has been modified. The conclusion part was expanded and the research prospect was discussed in depth.

“In addition, the evaluation standard of photocatalytic NRR reaction is usually based on the absolute yield and evolution rate of ammonia production (μmol·gcat-1 h-1 or μmol·h-1). With the innovation and development of the photocatalytic reaction system, reactors' design types have been enriched [92]. The application form of photocatalyst is no longer a single suspension type, and the supported catalyst that is convenient for recovery and utilization has also begun to receive attention [93]. How to make a uniform and fair comparison of the performance of these different types of photocatalysis systems has become a thorny problem. In addition to specifying various parameters in detail in the report, especially the number of substances that mainly play a catalytic role and corresponding active sites……

  1. Because of the low reaction efficiency caused by the poor solubility of the N2molecule, it is considered that the nitrogen source used in the photo(electro)catalytic reaction can be replaced. Nitrogen-based compounds like nitrate, nitrite, and nitrogen oxide are readily soluble in water. Therefore, the problem of N≡N cracking and activation could be avoided, and the hydrogen evolution reaction could be inhibited. Similarly, water vapor can also be used as the proton source. Simplifying the traditional gas-liquid-solid three-phase reaction into a gas-solid two-phase reaction is a potential method to improve the efficiency of the NRR reaction……

3.Recently, first-principle calculation combined with kinetic analysis has been widely used to predict the reaction potential barrier of the rate-determining step. However, the precise reaction kinetics theory has not been determined and is still developing. It is necessary to conduct more in-depth thermodynamic and kinetic studies to understand the catalytic performance of ammonia synthesis more practically.”

  1. There are some language errors and description errors in the article, which can be modified to make the article more beautiful.

Response: Thanks for the comment. The sentence grammar was checked and revised, and the article was polished. It was mainly marked in blue.

In this review manuscript, the authors summarized the main challenges and differences between the current photocatalytic and electrocatalytic nitrogen fixation systems. Based on the recent development on nitrogen fixation, the authors classified the modification strategies on photocatalysts and electrocatalysts. However, major revision is necessary before publication on Molecules. The comments are listed below:

  1. This manuscript focused much more on the photocatalysis in Chapter 3 while the challenges on electrocatalytic nitrogen fixation are absent.
  2. In Chapter 4, the modification strategies of on electrocatalysts including "Heterostructure construction" and "Cocatalyst addition" should also be added. In addition, many other modification strategies should be summarized systematically.
  3. The"4.3.2. Electrocatalyst" was miswritten as"4.1.2. Electrocatalyst" in the manuscript.
  4. The authors should polish the review manuscript carefully for language and grammar.

Reviewer 2 Report

The manuscript provides a review of the current state of the field of photoactivation for the electrochemical reduction of nitrogen to ammonia. The topic is of great importance as synthetic ammonia is required to produce food, possibly a means to store energy in the future.

The manuscript suffers a fair deal from inconsistent tense and other grammatical issues. For example, in the introduction, the past tense describes the current process of thermocatalytic synthesis. Incorrect tense is used throughout the paper.

The subheading format is awkward, with multiple subheadings having the same title.

The paper also uses many figures from previous papers with citations but without stating where the figures originate from or presumably permission from the journal and authors of the original work. See figures 1 and 4 as compared to DOI:10.1039/d1na00565k, 10.1021/acsami.9b16432

The review also suffers a great deal from a lack of details. In section 3, where the current challenges facing the field are discussed, there are 3 references, far too few to provide a detailed understanding. The issue arises in most sections, such as 4.1, where morphologic strategies are introduced in two sentences.

The review exclusively uses the phrase 'photocatalysis' while 'photolysis' is sometimes used to describe photodecomposition; though more limited in scope, these works shed light on the reaction. The difference is not important to the concept of the manuscript, but several studies related to this topic are overlooked because of the word choice. Specifically, works related to computational modeling.

Finally, the review is missing a critical section on theoretical modeling. Though uncommonly studied together, photolysis and electrochemistry are studied separately very often.

Author Response

Dear Editors and Reviewers,

Thanks for your comments concerning our manuscript entitled “Advances in Semiconductor-Based Nanocomposite Photocatalysts and Electrocatalysts for Nitrogen Reduction to Ammonia” (manuscript ID molecules-2247017). Those comments are all valuable and helpful for revising and improving our paper. We have carefully studied the comments and made corrections based on your comments. Modified portions are highlighted in red in the revised version. As suggested, we improved the manuscripts by improving the writing and adding more data and analyses. The revised manuscript is responded to, point by point.

The primary corrections in the paper and the responses to the reviewer’s comments are as follows.

Thank you and best regards.

Yours sincerely,

Corresponding author: Qian Su suqian93@163.com

Reviewer #2: 1.The manuscript suffers a fair deal from inconsistent tense and other grammatical issues. For example, in the introduction, the past tense describes the current process of thermocatalytic synthesis. Incorrect tense is used throughout the paper.

Response: Thanks for the comment. Ammonia was an essential component of modern chemical products and a building block for the molecules of life in nature [1]. The Haber-Bosch (H-B) process was mainly used for industrial-grade ammonia synthesis. Nitrogen was reduced to ammonia by hydrogen through a metal-based catalyst at 400-500 °C and 150-250 atm. However, the hydrogen used in this process was mainly from reforming hydrocarbons (CH4 + 2H2O = 4H2 + CO2). The release of large amounts of greenhouse gases accompanied the production. The H-B process was reported to consume 1-2% of the world’s total energy consumption and 3% of the global CO2 emissions [2].

In 1977, Schrauzer and Guth [3] found that Fe2O3-doped TiO2 powder could reduce nitrogen to ammonia under UV light irradiation. Subsequently, it was reported [4] that desert sands formed after weathering titanite-rich rocks could fix nitrogen by photoreaction in 1983. Tennakone’s group [5] prepared ultrafine Fe(O)OH particles with photocatalytic activity in reducing nitrogen to ammonia and discussed the preparation method, ammonia yield, and reaction mechanism of the catalyst in 1991. It confirmed the possibility of using solar energy to reduce nitrogen to synthesize ammonia directly and raised the curtain on the exploration and application of semiconductor materials in the field of photocatalytic nitrogen fixation. In recent years, stimulated by the goal of ‘double carbon’, the development of ecological priority and green artificial synthesis has been widely concerned by researchers. It combines photocatalytic technology with nitrogen fixation research, using sunlight as the driving energy and catalysts to reduce nitrogen into ammonia with high energy density, accessible storage, and transportation. In addition, it could convert light energy to chemical energy while achieving zero carbon emissions. The photocatalytic technology was used to reduce nitrogen into ammonia with high energy density and easy to store and transport by using catalysts. To implement the green development concept comprehensively and accurately, applying photocatalytic technology to the nitrogen cycle has become one of the hot spots of research in the new energy field.

  1. The subheading format is awkward, with multiple subheadings having the same title.

Response: Thanks for the comment. The title format has been changed in the manuscript

  1. The paper also uses many figures from previous papers with citations but without stating where the figures originate from or presumably permission from the journal and authors of the original work. See figures 1 and 4 as compared to DOI:10.1039/d1na00565k, 10.1021/acsami.9b16432.

Response: Thanks for the comment.Figures 1 and 4 have been revised in the manuscript

  1. The review also suffers a great deal from a lack of details. In section 3, where the current challenges facing the field are discussed, there are 3 references, far too few to provide a detailed understanding. The issue arises in most sections, such as 4.1, where morphologic strategies are introduced in two sentences.
  • Response:Thanks for the comment. 1. Low utilization of light energy

Quantum efficiency was an important parameter describing photoelectric conversion in photocatalysis technology. It refers to the ratio of the average number of photoelectrons produced in a specific wavelength per unit time to the number of incident photons. While, the quantum efficiency of the catalysts developed in recent years is usually lower than 5%, which was still at a low level [76-79].

76.Li CC, Wang T, Zhao ZJ, et al. Promoted fixation of molecular nitrogen with surface oxygen vacancies on plasmon-enhanced TiO2 photoelectrodes. Angewandte Chemie International Edition, 2018, 57(19): 5278-5282.

77.Ying Wang,Thomas J. Meyer. A route to renewable energy triggered by the Haber-Bosch process. Chem, 2019, 5(3): 496-497.

78.Miao Wang, Mohd A. Khan, Imtinan Mohsin, et al. Can sustainable ammonia synthesis pathways compete with fossil-fuel based Haber-Bosch processes? Energy & Environmental Science, 2021, 14: 2535-2548.

79.Jianping Guo, Ping Chen. Ammonia history in the making. Nature Catalysis, 2021, 4: 734-735.

3.2. Low separation rate of photogenerated carriers

How to suppress the migration of photogenerated carriers and reduce the complexation of electron-hole pairs is the focus of current research on photocatalysis [80-81].

80.Limin Yu, Mo Zhao, Xianlin Zhu, et al. Construction of 2D/2D Z-scheme MnO2-x/g-C3N4 photocatalyst for efficient nitrogen fixation to ammonia. Green Energy & Environment, 2021, 6(4): 538-545.

81.Liu Wenzhu, Sun Mingxuan, Ding Zhipeng, et al. Ti3C2 MXene embellished g-C3N4 nanosheets for improving photocatalytic redox capacity. Journal of Alloys and Compounds, 2021, 877, 160223.

3.3. N2 adsorption and activation difficulties

The reduction reaction of nitrogen in the presence of photocatalyst requires three steps: (1) dissolution of nitrogen in water; (2) diffusion of dissolved nitrogen in water to the surface liquid film of the catalyst; and (3) activation of nitrogen by adsorption on the active sites on the catalyst surface. However, the solubility of nitrogen in the aqueous phase reaction system at room temperature and pressure is extremely low (~1 mmol·L-1) and the expansion coefficient is small (10-5 cm2·s-1), resulting in the rate of ammonia synthesis being limited by the process of nitrogen dissolution and diffusion in the reaction system [82-83].

82.Hu KQ, Huang ZW, Zeng LW, et al. Recent advances in MOF-based materials for photocatalytic nitrogen fixation[J]. European Journal of Inorganic Chemistry, 2022, 2022(3), 202100748.

83.Amro M O Mohamed, Yusuf Bicer. The search for efficient and stable metal-organic frameworks for photocatalysis: atmospheric fixation of nitrogen. Applied Surface Science, 2022, 583, 152376.

  1. The review exclusively uses the phrase 'photocatalysis' while 'photolysis' is sometimes used to describe photodecomposition; though more limited in scope, these works shed light on the reaction. The difference is not important to the concept of the manuscript, but several studies related to this topic are overlooked because of the word choice. Specifically, works related to computational modeling.

Response: Thanks for the comment.

4.5. Computational modeling

The current computational modeling of photocatalytic and electrocatalytic reactions was mostly based on the density function theory (DFT) calculations. The DFT calculation by Tang et al. [84] revealed that in the MoS2-catalyzed NRR process, the positively charged edge Mo atom was the active site for activation of N2 molecules. The electrons could be transferred from the adsorbed N2 to the edge Mo. The electrons could be transferred from the adsorbed N2 to the edge Mo sites and form N-Mo bonds, thus weakening the N≡N bonds. Chu et al. [85] delved into the role of Mo doping in MnO2 catalysts and further evaluated the rationality of the distal binding mechanism by DFT calculations. With reference to the reported theoretical study and the fact that N2H4 was not found in the reaction products, a possible distal binding mechanism of Mo3/Fe2@ZC-cm photocatalyst during hydrogenation was proposed. In the nitrogen reduction reaction by the photocatalyst provides abundant photogenerated electrons, water provides protons (H+), and the metal sites near the oxygen vacancies could not only act as active centers to adsorb and activate N2 molecules, but also facilitate the charge transfer from the photocatalyst to the N2 molecules. In this environment, the Mo/Fe factor adsorption activates the distal N atom in the N2 molecule to bind to H+, weakening the stable N≡N triple bond. In the subsequent proton-coupled electron transfer process, the NH3 molecule was formed and successfully desorbed. The N atom of the other one continues to bind to H+ until another NH3 was formed and released. At this point, the reaction process of N2 reduction to NH3 was completed, and then the next cycle proceeds.

84.G. Gao, Y. Jiao, E. R. Waclawik, A. Du. Single atom (Pd/Pt) supported on graphitic carbon nitride as efficient photocatalyst for visible-light reduction of carbon dioxide. J. Am. Chem. Soc. 2016, 138, 6292.

85.Ke Chu, Ya-ping Liu, Yu-biao Li, et al. Multi-functional Mo-doping in MnO2 nanoflowers toward efficient and robust electrocatalytic nitrogen fixation. Applied Catalysis B: Environmental, 2020, 264: 118525.

  1. Finally, the review is missing a critical section on theoretical modeling. Though uncommonly studied together, photolysis and electrochemistry are studied separately very often.

Response: Thanks for the comment.

4.5. Computational modeling

The current computational modeling of photocatalytic and electrocatalytic reactions was mostly based on the density function theory (DFT) calculations. The DFT calculation by Tang et al. [84] revealed that in the MoS2-catalyzed NRR process, the positively charged edge Mo atom was the active site for activation of N2 molecules. The electrons could be transferred from the adsorbed N2 to the edge Mo. The electrons could be transferred from the adsorbed N2 to the edge Mo sites and form N-Mo bonds, thus weakening the N≡N bonds. Chu et al. [85] delved into the role of Mo doping in MnO2 catalysts and further evaluated the rationality of the distal binding mechanism by DFT calculations. With reference to the reported theoretical study and the fact that N2H4 was not found in the reaction products, a possible distal binding mechanism of Mo3/Fe2@ZC-cm photocatalyst during hydrogenation was proposed. In the nitrogen reduction reaction by the photocatalyst provides abundant photogenerated electrons, water provides protons (H+), and the metal sites near the oxygen vacancies could not only act as active centers to adsorb and activate N2 molecules, but also facilitate the charge transfer from the photocatalyst to the N2 molecules. In this environment, the Mo/Fe factor adsorption activates the distal N atom in the N2 molecule to bind to H+, weakening the stable N≡N triple bond. In the subsequent proton-coupled electron transfer process, the NH3 molecule was formed and successfully desorbed. The N atom of the other one continues to bind to H+ until another NH3 was formed and released. At this point, the reaction process of N2 reduction to NH3 was completed, and then the next cycle proceeds.

84.Zhang L, Ji X Q, Ren X, Ma Y J, Shi X F, Tian Z Q, Asiri A M, Chen L, Tang B, Sun X P. Boosted Electrocatalytic N2 Reduction to NH3 by Defect‐Rich MoS2 Nanoflower. Adv. Mater, 2018, 30. (28): 1800191.

85.Ke Chu, Ya-ping Liu, Yu-biao Li, et al. Multi-functional Mo-doping in MnO2 nanoflowers toward efficient and robust electrocatalytic nitrogen fixation. Applied Catalysis B: Environmental, 2020, 264: 118525.

Reviewer 3 Report

In this review manuscript, the authors summarized the main challenges and differences between the current photocatalytic and electrocatalytic nitrogen fixation systems. Based on the recent development on nitrogen fixation, the authors classified the modification strategies on photocatalysts and electrocatalysts. However, major revision is necessary before publication on Molecules. The comments are listed below:

1. This manuscript focused much more on the photocatalysis in Chapter 3 while the challenges on electrocatalytic nitrogen fixation are absent.

2. In Chapter 4, the modification strategies of on electrocatalysts including "Heterostructure construction" and "Cocatalyst addition" should also be added. In addition, many other modification strategies should be summarized systematically.

3. The "4.3.2. Electrocatalyst" was miswritten as "4.1.2. Electrocatalyst" in the manuscript.

4. The authors should polish the review manuscript carefully for language and grammar.

Author Response

Dear Editors and Reviewers,

Thanks for your comments concerning our manuscript entitled “Advances in Semiconductor-Based Nanocomposite Photocatalysts and Electrocatalysts for Nitrogen Reduction to Ammonia” (manuscript ID molecules-2247017). Those comments are all valuable and helpful for revising and improving our paper. We have carefully studied the comments and made corrections based on your comments. Modified portions are highlighted in red in the revised version. As suggested, we improved the manuscripts by improving the writing and adding more data and analyses. The revised manuscript is responded to, point by point.

The primary corrections in the paper and the responses to the reviewer’s comments are as follows.

Thank you and best regards.

Yours sincerely,

Corresponding author: Qian Su suqian93@163.com

Reviewer #3: 1. This manuscript focused much more on the photocatalysis in Chapter 3 while the challenges on electrocatalytic nitrogen fixation are absent.

Response: Thanks for the comment. 

3.5 Problems faced by electrocatalysis

      The research of electrocatalysis mainly includes the following aspects: 1) Combining with theoretical calculation, further improve the calculation method and model of NRR to provide theoretical guidance for catalyst design. 2) The occurrence of HER side reaction reduces the efficiency of electro-catalytic synthesis of NH3. 3) In the process of electrocatalysis, the structure of the intermediates produced by the reaction was complex and the time of production was short. It was difficult to characterize and capture these intermediates by existing experimental methods.

      The research of electro-catalytic reduction of nitrogen was very important for the development of ecological environment and new energy. The progress in the field of NRR has confirmed that it is possible to reduce N2 to NH3 using electrocatalysis at normal temperature and pressure. However, the efficiency of electrocatalysis is low. It is still in the laboratory research stage.

  1. In Chapter 4, the modification strategies of on electrocatalysts including "Heterostructure construction" and "Cocatalyst addition" should also be added. In addition, many other modification strategies should be summarized systematically.

Response: Thanks for the comment. 

At present, heterojunction electrocatalysts were mainly concentrated on type II catalysts. Hu et al. [86] have proved that compared with g-C3N4 or ternary metal sulfide, the construction of g-C3N4 with ternary metal sulfide could significantly improve the nitrogen fixation efficiency. When the mass fraction of ZnMoCdS was 80%, the heterojunction system has the highest nitrogen fixation capacity. A large number of sulfur vacancies not only act as active centers, but also greatly improve the electrocatalytic activity.

  1. Hu, Y. Li, F. Li, Z. Fan, H. Ma, W. Li, X. Kang, Construction of g-C3N4/Zn0.11Sn0.12Cd0.88S1.12 Hybrid Heterojunction Catalyst with Outstanding Nitrogen Photofixation Performance Induced by Sulfur Vacancies. ACS Sustainable Chem. Eng. 2016, 4, 2269.

Oshikiri et al. [87] doped the strontium titanate photoelectrode with Au NPs deposited on the surface by niobium immersed in the anode with ethanol as a sacrificial donor. On the other side Ru-based co-catalyst was used to modify Nb-SrTiO3 with N2-saturated aqueous HCl solution. The spacer of this photoelectrochemical system effectively prevented the further oxidation of ammonia to nitrate during the catalytic process. In addition, the chemical potential difference was generated by the different pH values of the cathode and anode chambers, thus accelerating the rate of NH3 synthesis.

Liu et al. [88] synthesized Co element-doped sulfur vacancy-rich MoS by introducing sulfur vacancies at the MoS basal position and using doped Co elements to replace Mo in the lattice. The catalyst showed excellent NRR activity. The best Faraday efficiency was 10% with an ammonia yield of 0.63 mmol·h-1·g-1. This was attributed to the fact that the Co element doping accelerated the adsorption and dissociation of N2 on the defective MoS, which in turn promoted the NRR process.

87.T. Oshikiri, K. Ueno, H. Misawa, Plasmon-Induced Ammonia Synthesis through Nitrogen Photofixation with Visible Light Irradiation. Angew. Chem., Int. Ed. 2014, 53, 9802.

88.Zhang J, Tian X Y, Liu M J, Guo H, Zhou J D, Fang Q Y LiuZ,Wu Q, Lou J. Cobalt-Modulated Molybdenum-Dinitrogen Interaction in MoS2 for Catalyzing Ammonia Synthesis. J. Am. Chem. Soc., 2019,141(49) : 19269.

4.5. Computational modeling

The current computational modeling of photocatalytic and electrocatalytic reactions was mostly based on the density function theory (DFT) calculations. The DFT calculation by Tang et al. [84] revealed that in the MoS2-catalyzed NRR process, the positively charged edge Mo atom was the active site for activation of N2 molecules. The electrons could be transferred from the adsorbed N2 to the edge Mo. The electrons could be transferred from the adsorbed N2 to the edge Mo sites and form N-Mo bonds, thus weakening the N≡N bonds. Chu et al. [85] delved into the role of Mo doping in MnO2 catalysts and further evaluated the rationality of the distal binding mechanism by DFT calculations. With reference to the reported theoretical study and the fact that N2H4 was not found in the reaction products, a possible distal binding mechanism of Mo3/Fe2@ZC-cm photocatalyst during hydrogenation was proposed. In the nitrogen reduction reaction by the photocatalyst provides abundant photogenerated electrons, water provides protons (H+), and the metal sites near the oxygen vacancies could not only act as active centers to adsorb and activate N2 molecules, but also facilitate the charge transfer from the photocatalyst to the N2 molecules. In this environment, the Mo/Fe factor adsorption activates the distal N atom in the N2 molecule to bind to H+, weakening the stable N≡N triple bond. In the subsequent proton-coupled electron transfer process, the NH3 molecule was formed and successfully desorbed. The N atom of the other one continues to bind to H+ until another NH3 was formed and released. At this point, the reaction process of N2 reduction to NH3 was completed, and then the next cycle proceeds.

  1. Zhang L, Ji X Q, Ren X, Ma Y J, Shi X F, Tian Z Q, Asiri A M, Chen L, Tang B, Sun X P. Boosted Electrocatalytic N2Reduction to NH3 by Defect‐Rich MoS2  Adv. Mater, 2018, 30. (28): 1800191.
  2. Ke Chu, Ya-ping Liu, Yu-biao Li, et al. Multi-functional Mo-doping in MnO2 nanoflowers toward efficient and robust electrocatalytic nitrogen fixation. Applied Catalysis B: Environmental, 2020, 264: 118525.
  3. The "3.2. Electrocatalyst" was miswritten as "4.1.2. Electrocatalyst" in the manuscript.

Response: Thanks for the comment. 4.3.2. Electrocatalyst

  1. The authors should polish the review manuscript carefully for language and grammar.

Response: Thanks for the comment. Revised in the manuscript

Round 2

Reviewer 1 Report

The paper can be accepted in its present form.

Reviewer 2 Report

The Authors have made significant improvements to the manuscript, and I believe it is suitable for publication. There are minor grammatic issues that a line editor will surely correct.

Reviewer 3 Report

The authors have revised the manuscript according to the comments and I suggest that this work could be accepted without further revision.